# Regulating Macrophages through Immunomodulatory Biomaterials Is a Promising Strategy for Promoting Tendon-Bone Healing

**DOI:** 10.3390/jfb13040243

**Published:** 2022-11-15

**Authors:** Haihan Gao, Liren Wang, Haocheng Jin, Zhiqi Lin, Ziyun Li, Yuhao Kang, Yangbao Lyu, Wenqian Dong, Yefeng Liu, Dingyi Shi, Jia Jiang, Jinzhong Zhao

**Affiliations:** 1Department of Sports Medicine, Shanghai Sixth People’s Hospital Affiliated to Shanghai Jiao Tong University School of Medicine, Shanghai 200233, China; 2Regenerative Sports Medicine and Translational Youth Science and Technology Innovation Workroom, Shanghai Jiao Tong University School of Medicine, Shanghai 200025, China; 3Regenerative Sports Medicine Lab of the Institute of Microsurgery on Extremities, Shanghai Sixth People’s Hospital Affiliated to Shanghai Jiao Tong University School of Medicine, Shanghai 200233, China

**Keywords:** macrophage, tendon-to-bone interface, tendon-bone healing, regenerative medicine, biomaterials

## Abstract

The tendon-to-bone interface is a special structure connecting the tendon and bone and is crucial for mechanical load transfer between dissimilar tissues. After an injury, fibrous scar tissues replace the native tendon-to-bone interface, creating a weak spot that needs to endure extra loading, significantly decreasing the mechanical properties of the motor system. Macrophages play a critical role in tendon-bone healing and can be divided into various phenotypes, according to their inducing stimuli and function. During the early stages of tendon-bone healing, M1 macrophages are predominant, while during the later stages, M2 macrophages replace the M1 macrophages. The two macrophage phenotypes play a significant, yet distinct, role in tendon-bone healing. Growing evidence shows that regulating the macrophage phenotypes is able to promote tendon-bone healing. This review aims to summarize the impact of different macrophages on tendon-bone healing and the current immunomodulatory biomaterials for regulating macrophages, which are used to promote tendon-bone healing. Although macrophages are a promising target for tendon-bone healing, the challenges and limitations of macrophages in tendon-bone healing research are discussed, along with directions for further research.

## 1. Introduction

The tendon-to-bone interface is the connecting structure between the tendon and bone in the motor system of the human body [1]. The length of the tendon-to-bone interface ranges from 200 μm to 1 mm [2]. Despite its microscopic size, the tendon-to-bone interface is an indispensable structure for maintaining the normal function of the motor system [2]. The elasticity modulus of tendons is 200 MPa, while that of bones is 20 Gpa [3]. Due to increased stress concentrations, the junction interface of two dissimilar materials with significantly mismatched mechanical properties is a weak spot when enduring extra loading [4]. However, injuries to the motor system usually occur on the bone or tendon side, rather than at the tendon-to-bone interface, due to its intricate structure [5]. Structurally, the tendon-to-bone interface can be histologically divided into four layers: tendon, uncalcified fibrocartilage, calcified fibrocartilage, and bone [2]. These four layers allow for a smooth stress transfer from the tendon to the bone while avoiding stress concentrations at the interface [2]. The clinical gold standard treatment for tendon-to-bone injury (e.g., a rotator cuff tear) is surgically reattaching the remnant tendon to its footprint [1]. Although surgery can restore the continuity from tendon to bone, fibrous scar tissue will replace uncalcified and calcified fibrocartilage at the tendon-to-bone interface after healing [3,6]. Fibrous scar tissue cannot effectively distribute the stress concentrations at the tendon-to-bone interface, which subsequently creates retear potential at the injury site after healing [3]. The retear rate after rotator-cuff surgical repair can reach 94%, which is closely related to the significantly reduced mechanical strength of the fibrous scar after surgery, compared to that of the native structure [7]. Current research promoting tendon-bone healing focuses on preventing the formation of fibrous scar tissue and encouraging the regeneration of the native tendon-to-bone interface.

Mounting evidence suggests that the immune system, particularly macrophages, plays a critical role in the regeneration of various tissues [8,9]. According to their functions and surface markers, macrophage phenotypes can be divided into M1 pro-inflammatory and M2 anti-inflammatory macrophages [10]. M1 macrophages arise during the early stages following tissue injury, the function of which lies in phagocytosing pathogens and cellular debris and secreting inflammatory cytokines (e.g., IL-1 and TNF-α), to exacerbate the inflammation [11]. After the acute inflammation phase, M2 macrophages accumulate during tissue repair and regeneration and influence the tissue regeneration processes by secreting IL-4, IL-13, and TGF-β [11]. An imbalance between M1 and M2 macrophages results in imperfect regeneration, while the tissue regeneration processes can be facilitated by regulating the macrophage phenotypes through chemical, physical, or biological therapies [12,13]. Until now, it has been uncertain exactly how macrophages exert an influence on tendon-bone healing. Hence, more attention should be paid when modulating macrophage phenotypes during tendon-bone healing, to avoid undesirable inflammatory responses when promoting tendon-bone healing.

In this review, the structure and function of the tendon-to-bone interface are introduced, followed by a review of the recent advances regarding the role of different macrophage phenotypes in the tendon-bone healing processes. Then, current studies on enhancing tendon-bone healing through various immunomodulatory biomaterials modifying the macrophage phenotypes are presented (Figure 1). Finally, the challenges and limitations of macrophages in tendon-bone healing research are summarized and future research directions are discussed.

## 2. Structure and Function of the Tendon-to-Bone Interface

The connection between the tendon/ligament and bone can be divided into fibrous tendon-to-bone interfaces and fibrocartilage tendon-to-bone interfaces [14]. The deltoid muscle attaches to the humerus across the fibrous tendon-to-bone interface, where the tendon is directly connected to the periosteum with Sharpey fibers and is capable of dispersing longitudinal tension and lateral shear forces [3]. Conversely, fibrocartilaginous tendon-to-bone interfaces are more common and are found at the end of the rotator cuff, anterior cruciate ligament, and Achilles tendon, which are closely related to clinical diseases of great significance [14]. Hence, this review focuses on fibrocartilage tendon-to-bone interfaces.

Despite its tiny size, the fibrocartilage tendon-to-bone interface constitutes four zones. The first layer is the tendon/ligament tissue, formed primarily of fibroblasts, while the extracellular matrix is mainly composed of parallel organized type-I collagen fibers [2,15]. The second layer is uncalcified fibrocartilage, which consists primarily of fibrocartilage cells arranged in columns along the direction of tendon tension, and the extracellular matrix, composed mostly of type-I collagen, type-II collagen, and proteoglycans [2,16]. The third layer is calcified fibrocartilage, which is mostly made up of hypertrophic fibrocartilage cells, and an extracellular matrix composed of type-II and type-X collagen [2,16]. The fourth layer is of bone, generally composed of osteocytes, osteoblasts, and osteoclasts, with the extracellular matrix containing a high proportion of mineralized type-I collagen [2].

Although the fibrocartilage tendon-to-bone interface constitutes four layers, it is important to note that the four layers are structurally continuous, despite noticeable tissue variances in extracellular matrix composition. A three-dimensional reconstruction of the direction of fibers at the tendon-to-bone interface of the Achilles tendon revealed that the fibers were continuous from the tendon to the bone side [17]. However, the cross-sectional areas gradually decreased, while fiber curvature increased [17]. The stress would peak, theoretically, at the tendon-to-bone interface as a result of observed fiber alignment, implying that there may be a unique mechanism to reduce stress concentration at the tendon-to-bone interface [17,18]. Using a customized confocal microscopy–mechanical loading system, Rossetti et al. [2] found a transitional zone of 500 μm in the tendon-to-bone interface. Tendon fibers with a diameter of 100 nm gradually unravel into interface fibers with a diameter of about 30 nm in this zone, with the fiber orientation shifting from parallel in the tendon to random orientation in the bone [2]. When the tendon-to-bone interface is subjected to non-axial stress, this structure, with its load-sharing mechanism, ensures that multiple interface fibers are tensed simultaneously, which reduces the stress load on a single fiber and significantly increases the maximum load that the tendon-to-bone interface can withstand [2]. 

Clinical studies have revealed that after the surgical reattachment of the ruptured tendon to the bone, the regenerated interface fails to form the native four-layer structure, instead forming fibrous scar tissue in its place [3,6] (Figure 2). Due to the irregular fiber organization of the extracellular matrix of fibrous scar tissue, which lacks a load-sharing mechanism, the maximal load of the tendon-to-bone complex significantly decreases after injury, leading to an increased rate of postoperative retearing [3]. The intricate structure of the tendon-to-bone interface lays the foundation for its mechanical properties [19]. Until now, the major issue facing tendon-to-bone healing has entailed the induction of the perfect regeneration of the native tendon-to-bone interface structure.

## 3. Role of Macrophages in Tendon-Bone Healing

Tendon-bone healing comprises three stages: the inflammatory stage, the repair stage, and the remodeling stage [20,21]. Macrophages appear in substantial populations in the early stages of tendon-bone healing and persist throughout the healing process, regulating cell proliferation, differentiation, and extracellular matrix formation by secreting various inflammatory factors and growth factors [22,23]. The importance of macrophages in tendon-bone healing has been progressively emphasized in recent years, and the processes by which they affect tendon-bone healing have gradually been uncovered.

### 3.1. Macrophages Phenotypes and Spectrum

Macrophages can be categorized into M1 and M2 phenotypes, based on their pro-inflammatory and anti-inflammatory functions [24,25]. M1 macrophages are primarily responsible for the eradication of harmful bacteria and the stimulation of inflammatory reactions, while M2 macrophages are responsible for the regulation of immunosuppression or allergic responses [24,25]. Categorizing macrophages as merely M1 or M2 is simplistic and does not accurately reflect the heterogeneity of macrophages in vivo; besides, it does not allow for in-depth research into the role that they play in tissue regeneration and repair. It is currently suggested that macrophage phenotypes can be further characterized, based on inducing stimuli; this is widely accepted and is used to classify macrophages in terms of tissue regeneration and repair [26]. The phenotypes of macrophages can be influenced by different stimuli, and, subsequently, the induced phenotypes of macrophages are capable of reacting in terms of tissue regeneration and repair (Figure 3). Under the stimulation of TNF-α, IFN-γ, and lipopolysaccharide (LPS), M0 macrophages can polarize to M1 macrophages; these clear pathogenic microorganisms and cell debris, while producing high levels of reactive oxygen species (ROS) and inflammatory factors (e.g., IL-1β and IL-6) [27]. Macrophages can convert into the M2a type, in response to IL-4 and IL-13 stimulation, and release arginase-1 (Arg-1) and insulin-like growth factor-1 (IGF-1) to enhance vascularization and collagen formation. M2a macrophages are essential in the synthesis of an extracellular matrix, which is indispensable for tissue repair [27,28]. The formation of M2b macrophages can be induced by immune complexes and LPS and will express CD86, CD68, and MHCII, which can secrete IL-10 to suppress inflammation and produce high levels of inflammatory factors (e.g., TNF-α, and IL-1β) [29,30]. The M2c macrophages, which can be induced by glucocorticoids, IL-10, and TGF-β, are capable of secreting IL-10 and matrix metalloproteinase, to regulate extracellular matrix remodeling and fibrosis [27,31]. The formation of M2d macrophages can be induced by IL-6 and adenosine, which secrete transforming growth factor β (TGF-β) and vascular endothelial growth factor (VEGF) to enhance granulation tissue formation [32]. Until now, the specific signaling and differentiation cascades that produce specific macrophage phenotypes have remained unknown, and the significance of different phenotypes in the context of tissue regeneration has not yet been fully elucidated.

### 3.2. The Impact of M1 Macrophages on Tendon-Bone Healing

M0 macrophages in circulation, bone marrow, and synovial membrane are recruited to the tendon-to-bone interface and polarize primarily into M1 macrophages during the inflammatory stage [22]. The expression levels of M1 macrophage markers at the tendon-to-bone interface peak on the third day after a rotator cuff injury and subsequently decrease gradually [33]. M1 macrophages can phagocytize cellular debris and eliminate foreign pathogens, as well as influence tendon-bone healing by secreting inflammatory factors (e.g., IL-1β, TNF-α, and IL-6). 

IL-1β is rarely detected at native tendon-to-bone interfaces, but it is abundant in the fibrous scar tissue at the tendon-to-bone interface and in the surrounding joint fluid, which is assumed to be associated with poor tendon-bone healing [34,35]. IL-1β significantly reduces the expression of type-I collagen mRNA in tendon-derived progenitor cells and promotes the expression of cyclooxygenase-2, MMP1, MMP3, and prostaglandin E2 in human tendon cells, which correlates with the degradation of extracellular matrices [36,37]. Moreover, IL-1β impairs tendon-bone healing via the NF-κB pathway [38]. It activates the IκB kinase (IKK) complex, which phosphorylates and degrades the NF-κB dimer inhibitor, IκB, enabling NF-κB dimer transfer into the nucleus to increase the pro-inflammation factor gene expression [39]. The increased expression of IKK and NF-κB is detected in the tendon tissues of rotator cuff-tear patients and the tendon-to-bone interface tissues of acute rotator cuff-tear mice [38]. In IKK constitutively active mice, increased CD68^+^ macrophage infiltration and an apparent degeneration were observed in the tendon-to-bone interface, leading to a loss of the metachromasia at the fibrocartilage layer in the tendon-to-bone interface, and less bone volume in the humeral head [38]. The regeneration process after a tendon-to-bone interface injury was significantly improved in IKK conditional knock-out mice, compared to wild-type or IKK constitutively active mice, suggesting that activated NF-κB pathway impairs tendon-bone healing and that these changes may be related to an increase in pro-inflammatory cytokines and M1 macrophages (Figure 4) [38]. When the NF-κB pathway is activated, it leads to the decreased expression of Sox9, hindering the chondrogenic differentiation of MSCs, which may contribute to the poor regeneration of the fibrocartilage layer during tendon-bone healing [40,41]. IL-1β also leads to the excessive activity of osteoclasts around the tendon-to-bone interface, resulting in bone loss during tendon-bone healing and poor regeneration of the tendon-to-bone interface [42,43,44]. Hence, it can be concluded that IL-1β impedes tendon-bone healing, either directly or indirectly, by activating the NF-κB pathway, an important pathway that is implicated in poor tendon-to-bone interface regeneration.

TNF-α is also an inflammatory factor that is produced substantially by M1 macrophages during tendon-bone healing [45]. TNF-α expression is considerably higher in torn tendons, following rotator-cuff damage, than in intact tendons, which leads to the increased apoptosis of tendon stem cells, increased matrix metalloproteinase synthesis, and decreased type-I collagen synthesis, resulting in the degradation of the extracellular matrix [46,47]. TNF-α can also activate the NF-κB pathway, leading to the large-scale production of pro-inflammatory factors that exacerbate the inflammatory response during tendon-bone healing [39]. Under the stimuli of TNF-α, monocytes can be induced into forming osteoclasts, resulting in bone loss during tendon-bone healing [48]. Reducing TNF-α levels in Lewis rats decreases the number of M1 macrophages during tendon bone-healing and stimulates the regeneration of the fibrocartilage layer, ultimately improving biomechanical strength [49].

IL-6 is a downstream product of the NF-κB pathway and is abundant in torn tendons after rotator cuff tears, wherein it may play a dual role in tendon-bone healing [50,51]. IL-6 has both pro-inflammatory and anti-inflammatory properties, with the membrane-bound IL-6R mediating its anti-inflammatory effects and the soluble IL-6R mediating its pro-inflammatory effects [52]. While IL-6 can recruit monocytes and promote their differentiation into osteoclasts, they also enhance the release of the IL-1 receptor antagonist and IL-10 to suppress inflammation [50,52]. The level of TNF-α was higher in IL-6 knock-out mice than in wild-type mice after a patellar tendon injury, with the ratio of type-III collagen to type-I collagen in their injured tendons also being increased, suggesting that IL-6 is involved in the inflammation process and the remodeling of extracellular matrices during tendon repair [53]. Surprisingly, however, biomechanical tests showed that the maximum stress of an injured tendon was lower in IL-6 knock-out mice than in wild-type mice [53]. The above findings indicate that the role of IL-6 in tissue regeneration is complicated, with the influence of IL-6 on tendon-bone healing needing further research.

M1 macrophages not only regulate tissue regeneration by secreting inflammatory factors but they also release chemokines (e.g., CCL2, CXCL8, and SDF-1) to recruit mesenchymal stem cells (MSCs) to the site of damage [54,55]. After a rotator cuff injury, bone marrow MSCs can infiltrate through the bone tunnel on the greater tuberosity of the humerus to the tendon-to-bone interface, to promote tendon-bone healing [56]. Yang et al. (2019) used second near-infrared fluorescence imaging to track the distribution of exogenous MSCs in mice. After a supraspinatus tendon tear, exogenous MSCs were injected into the joint cavity immediately and appeared in large numbers at the injured tendon-to-bone interface on day 3 after injection, which facilitated the tendon-bone healing [57]. All these findings demonstrate that endogenous or exogenous MSCs can be recruited to the tendon-to-bone interface to facilitate tendon-bone healing. The chemokines released by M1 macrophages may play an essential role in MSCs recruitment during tendon-bone healing.

In general, during the early stages of tendon-bone healing, a large infiltration of M1 macrophages can phagocytize the cellular debris and foreign pathogens, while also recruiting MSCs to the tendon-to-bone interface by secreting chemokines, which has a beneficial effect on tendon-bone healing. Regrettably, once the inflammatory factors are secreted in excess, the inflammatory response of the local microenvironment will be exacerbated, which is detrimental to tendon-bone healing.

### 3.3. The Impact of M2 Macrophages on Tendon-Bone Healing

M1 macrophages account for the majority of macrophages during the inflammatory phase of tendon-bone healing, while M2 macrophages that are characterized by CD206 and Arg-1 are predominant during the subsequent repair and remodeling stages [22,33,58]. Due to the plasticity of macrophages, M2 macrophage accumulation during tissue healing may be derived from the repolarization of M1 macrophages or the induction of M0 macrophages under IL-4, IL-13, and other stimuli [59].

The ability of M2 macrophages to phagocytose pathogens and produce pro-inflammatory factors is limited, compared to that of the M1 macrophages, while their ability to release anti-inflammatory factors, including IL-4 and IL-10, is significantly increased [33,60]. These anti-inflammatory factors provide a suitable regenerative environment for subsequent tissue regeneration [61]. An M2 macrophage-conditioned medium promoted in vitro MSC osteogenesis; nevertheless, this positive effect was diminished with the addition of IL-10 neutralizing antibodies, suggesting that IL-10 is an important factor for M2 macrophages, to regulate tissue regeneration [62].

M2 macrophages also produce growth factors, such as TGF-β and VEGF, to regulate tissue regeneration [63,64]. TGF-β is highly attractive to macrophages and can form a positive feedback regulation of TGF-β and other growth factors via stimulating the macrophages [65,66]. The isoforms and concentrations of TGF-β play a decisive role in scar formation and tissue regeneration [67,68,69]. During tendon-bone healing after an acute rotator cuff injury in rats, TGF-β1 reaches the peak level at 10 days and is abundant in fibrous scar tissue, the main component of which is type-III collagen, indicating that TGF-β1 may be responsible for fibrous scar tissue formation at the injured tendon-to-bone interface [70]. When using exogenous TGF-β1 to promote tendon-bone healing after an acute rotator cuff injury in a rat model, although biomechanical strength was improved, TGF-β1 can elevate the transcriptional level of Col3a1 and enhance the formation of fibrous scar tissue [71]. In contrast, TGF-β3, which promotes fibrocartilage formation, is rarely expressed during tendon-bone healing [70,72]. Histological analysis revealed that the addition of TGF-β3 promoted the regeneration of fibrocartilage at the injured tendon-to-bone interface in the rotator cuff, thereby enhancing biomechanical strength after healing, with limited scar formation [73].

The growth factors released by macrophages, such as VEGF and BMP-2, play a significant role in tissue regeneration as well [74,75]. M2 macrophages were induced by sulfated chitosan in a mouse model, which subsequently facilitated endogenous VEGF production to induce the vascularization of ischemic disease [76]. Bone marrow MSC-derived exosomes stimulate the polarization of M2 macrophages and increase the expression of VEGF, which promotes vascularization around the injured tendon-to-bone interface in rats, beneficial for tendon-bone healing [77]. The regulation of M2 macrophage polarization through a surface topography design of honeycomb-like TiO_2_ can facilitate macrophages to release more BMP-2, to promote osteogenesis in a rat tibia implantation model [78].

M2 macrophages are believed to gradually replace M1 macrophages after the inflammatory stage to regulate tissue regeneration by secreting various anti-inflammatory and growth factors. Although the evidence that M2 macrophages promote tendon-bone healing by suppressing the inflammatory response is quite strong, the role of growth factors secreted by M2 macrophages on tendon-bone healing remains vague and controversial. For example, TGF-β1 secreted by macrophages may be the main cause of fibrous scar formation. Further studies are needed to illustrate the intricated correlation between the various factors secreted by macrophages and tendon-bone healing.

## 4. Immunomodulatory Biomaterials for Promoting Tendon-Bone Healing

### 4.1. Exosomes and Secretome

Exosomes are lipid bilayer membrane vesicles released by living cells. MSC-derived exosomes can regulate the phenotype of macrophages, thereby indirectly influencing tissue regeneration [79,80]. Xu et al. (2022) injected infrapatellar fat-pad MSC-derived exosomes into the bone tunnel after anterior cruciate ligament (ACL) reconstruction in rats and found that these exosomes promoted the polarization of M2 macrophages at the tendon-to-bone interface, along with bone tissue regeneration around the graft, improving the biomechanical strength of the graft at 4 and 8 weeks, compared to that in the non-exosome-treated group [81]. Shi et al. (2020) used bone marrow MSC-derived exosomes to promote tendon-bone healing in mouse Achilles tendons and found that bone marrow MSC-derived exosomes down-regulated the expression of inducible nitric oxide synthase (iNOS) and increased the expression of Arg-1 [82]. Compared to that in the control group without exosomes, more fibrocartilage regeneration at the tendon-to-bone interface and better biomechanical strength recovery were observed in the exosome group [83]. In other research, exosomes derived from mesenchymal stem cells were injected via the tail vein for rotator-cuff repair in a rat model [77]. Here, it was found that the exosomes decreased the expression of CD86 (an M1 macrophage marker) at the tendon-to-bone interface after a rotator cuff injury and suppressed the levels of IL-1β and TNF-α during tendon-bone healing. This promoted fibrocartilage regeneration and improved biomechanical strength [77] (Figure 5).

In addition to the exosomes, the secretome of MSCs and platelet-rich plasma may also influence tendon-bone healing by regulating macrophage polarization [83,84]. Using gelatin sponges to load human MSCs-derived conditioned medium (hBMSC-CM), Chen et al. (2021) discovered that hBMSC-CM decreased the expression of CD86 and increased the expression of CD163 (an M2 macrophage marker) during rat rotator cuff tendon-bone healing, compared to DMEM medium alone [83]. When the macrophages were cleared by liposomal clodronate, the effect of hBMSC-CM on tendon-bone healing was attenuated, showing that hBMSC-CM improves tendon-bone healing by means of modulating macrophage polarization [83]. This study also indicated that the regulation of macrophage polarization by hBMSC-CM may be mediated by smad2/3 phosphorylation since suppressing smad2/3 phosphorylation by SB431542 significantly reduced the inductive effect of the M2 macrophages by hBMSC-CM [83] (Figure 6). Comparing the effects of leukocyte-rich platelet-rich plasma and leukocyte-poor platelet-rich plasma on tendon-bone healing in a mouse rotator-cuff repair model, it was discovered that few M2 macrophages were observed in the tendon-to-bone interface at 8 weeks in the control group without PRP treatment, while more M2 macrophages were observed in the tendon-to-bone interface in both PRP-treated groups at 8 weeks [85]. These results illustrate that multiple sources of exosomes and the secretome can promote the polarization of M2 macrophages to facilitate tendon-bone healing.

### 4.2. Immunomodulatory Molecules

Various small molecules can modulate inflammatory responses and regulate macrophage polarization during tendon-bone healing through tissue engineering or systemic administration to impact the healing outcome [75,86]. It was found that an electrospun membrane, fabricated with PCL and melatonin, had an immunomodulatory effect to enhance rats’ rotator cuff tendon-bone healing [86]. This electrospun membrane could release melatonin steadily and decrease macrophage infiltration during tendon-bone healing, which promoted chondroid zone formation and decreased fibrovascular tissue formation [87] (Figure 7).

Wang et al. (2021) delivered acetylcholine and pyridostigmine, which have anti-inflammatory effects, through a fibrin gel to the tendon-to-bone interface after mouse rotator-cuff injury, which promoted fibrocartilage-like tissue regeneration during the healing stage [88]. Wang et al. (2021) wrapped a magnesium-pretreated periosteum around an auto-tendon graft for rabbit ACL reconstruction and confirmed that a magnesium-pretreated periosteum promoted the polarization of the M2 macrophage around the tendon-to-bone interface between the graft and the bone tunnel, compared with a stainless steel-pretreated periosteum, which is capable of preventing bone loss around the tunnel and increasing the maximum load to failure [89]. Chondroitin sulfate and bone morphogenetic protein-2 were attached to the surface of a polyethylene terephthalate (PET) ligament via polydopamine [90,91]. The chondroitin sulfate on the PET ligament increased the expression of Arg-1 and exerted a synergistic effect, with bone morphogenetic protein-2 used for bone regeneration during tendon-bone healing in a rat proximal-tibia graft-to-bone healing model [90] (Figure 8). Disulfiram can reduce the release of IL-1β and TNF-α from macrophages while promoting the M1 macrophage polarization to M2 macrophages [92]. Disulfiram reduces macrophage infiltration, inhibits peritendinous fibrosis, and promotes the regeneration of fibrocartilage and bone during tendon-bone healing after a mouse model Achilles tendon injury [92]. It is, therefore, feasible and promising to regulate tendon-bone healing by regulating macrophage polarization, using molecules that have immunomodulatory effects.

## 5. Conclusions and Perspectives

The significance of macrophages and inflammatory responses in tissue regeneration has garnered increasing attention in recent years and has been recognized as a potential and promising target for accelerating tissue regeneration [93,94]. This article provides a summary of the present state of research regarding the involvement of macrophages in tendon-bone healing and the promotion of tendon-bone healing by modulating macrophage polarization (Table 1). Despite the fact that several studies have demonstrated that macrophages are a promising target for promoting tendon-bone healing, numerous obstacles remain.

The first problem is the inconsistent evidence presented by researchers. The regenerative and immune capacities are negatively correlated since the development of immune competence is connected to a decline in regenerative potential [8]. In studies concerning tendon-bone healing, small animals such as mice, rats, and rabbits are the most commonly used animal models, with their regenerative capacities being substantially better than those of dogs, sheep, and humans [95,96,97]. In light of this fact, the findings obtained in studies conducted on small animal models may not necessarily be consistent with the findings obtained in studies on large animals or humans. This is an unavoidable limitation that must be taken into account when researching the function of macrophages in tendon-bone healing, and care must be taken when interpreting the experimental results. Although the results of experiments performed on large animals may be more comparable to humans, the higher cost of large animals for testing and the restrictions on antibodies make it more challenging to investigate the mechanisms in greater depth. Therefore, exploring the relevant mechanisms in small animal models and validating the results in large animal models may be a more appropriate experimental approach, to further investigate the function of macrophages in tendon-bone healing.

In addition to M1 and M2 phenotypes, macrophages can be further subdivided into several different phenotypes, each performing a unique function and playing a unique part in the process of tissue regeneration. However, the current research on tendon-bone healing merely classifies macrophages into M1 and M2 phenotypes, which simplifies the study but is not conducive to an in-depth study on the role of different macrophage phenotypes in tendon-bone healing. For this reason, the synergetic effects of various pro-inflammatory factors and the growth factors secreted by macrophages, as well as the time points of the appearance of different macrophage phenotypes during tendon-bone healing, should be researched in the future.

It has been widely proven that regulating macrophage phenotypes can promote tendon-bone healing, but how, and in what manner, remains elusive. To promote tendon-bone healing, additional research is required to elucidate the roles played by the various phenotypes of the macrophages. When compared to oral anti-inflammatory medications (e.g., indomethacin and celecoxib), the delivery of immunomodulatory substances that can modulate local macrophage phenotypes through tissue engineering can promote tendon-bone healing, while simultaneously reducing the risk of systemic adverse effects [75,98]. Thus, the precise modulation of macrophages by tissue engineering will be a promising means for promoting tendon-bone healing in the future.

In addition to macrophages, other immune cells such as T and B cells from the adaptive immune system also play an important role in tissue regeneration and repair [9,99,100]. The activity of T helper cells is also required for the polarization of M1 and M2 macrophages [11]. Therefore, it will be necessary for future studies to investigate whether other immune cells influence tendon-bone healing through macrophages or whether they have a regulatory effect on tendon-bone healing.

In conclusion, macrophages represent an important prospective target for promoting tendon-bone healing. Nevertheless, the ongoing research still faces several obstacles. In the future, research should concentrate on exploring the mechanisms of the macrophages regulating tendon-bone healing and on designing tissue engineering strategies to precisely modulate macrophage phenotypes for tendon-to-bone interface regeneration.

## Figures and Tables

**Figure 1 jfb-13-00243-f001:**
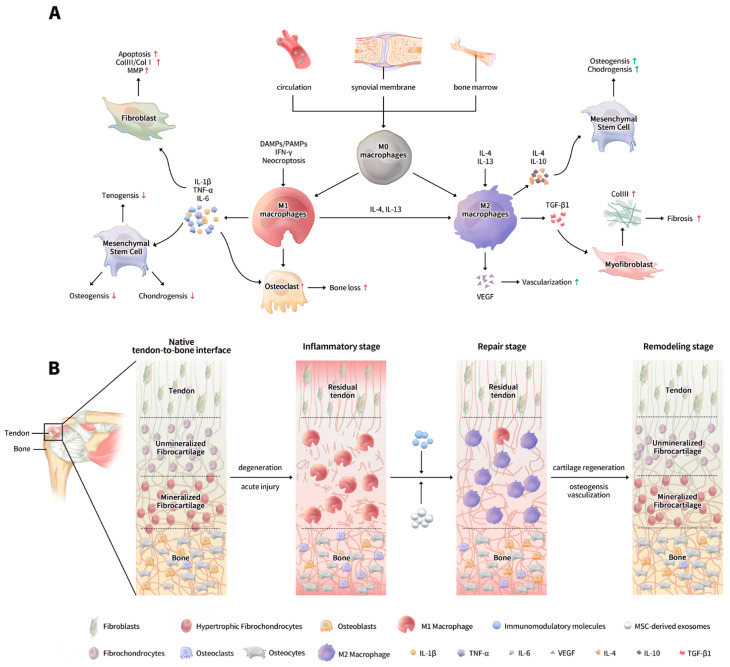
The function and regulation of macrophages during tendon-bone healing. (**A**) Macrophages are recruited to the tendon-to-bone interface from the circulation, the synovial membrane, and bone marrow. Under a specific stimulus, M0 macrophages can polarize to M1 and M2 macrophages, which then release pro-inflammatory factors, anti-inflammatory factors, and growth factors to influence the fibroblasts, mesenchymal stem cells, osteoclasts, and myofibroblasts; (**B**) M1 macrophages are predominant in the inflammatory stage. Using MSC-derived exosomes and immunomodulatory molecules can repolarize the M1 macrophages to M2 macrophages, which promotes tendon-bone healing. DAMP: damage-associated molecular pattern; PAMP: pathogen-associated molecular pattern; IFN: interferon; MMP: matrix metalloproteinases; IL: interleukin; TNF: tumor necrosis factor; TGF: transforming growth factor; VEGF: vascular endothelial growth factor; MSC: marrow stem cell.

**Figure 2 jfb-13-00243-f002:**
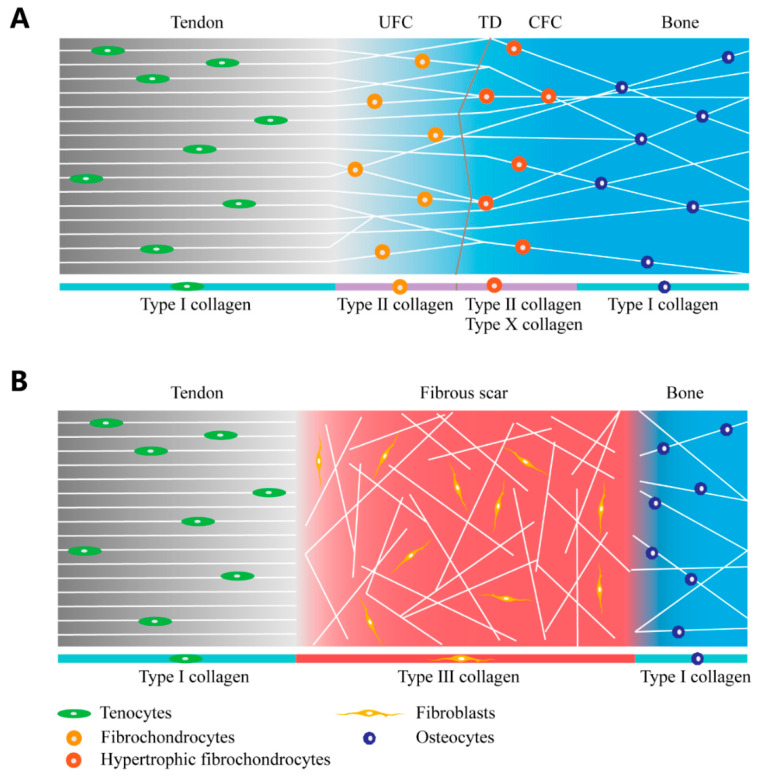
The native structure of the tendon-to-bone interface and fibrous scar tissue of the injured tendon-to-bone interface. (**A**) Schematic of the native tendon-to-bone interface; (**B**) schematic of fibrous scar tissue in an injured tendon-to-bone interface. UFC: uncalcified fibrocartilage; CFC: calcified fibrocartilage; TD: tidemark.

**Figure 3 jfb-13-00243-f003:**
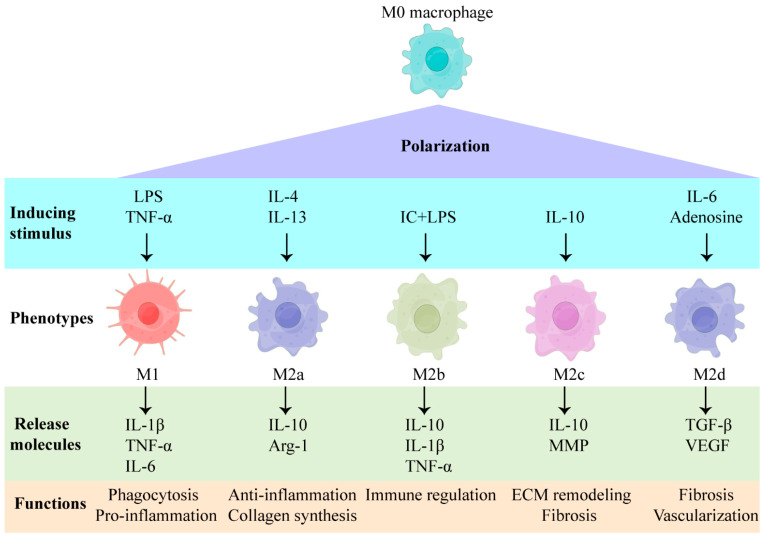
Inducing a stimulus and the factors secreted by macrophages of different phenotypes. LPS: lipopolysaccharide; TNF-α: tumor necrosis factor-alpha; IL: interleukin; IC: immune complex; MMP: matrix metalloproteinase; Arg-1: arginase-1; TGF-β: transforming growth factor beta; VEGF: vascular endothelial growth factor.

**Figure 4 jfb-13-00243-f004:**
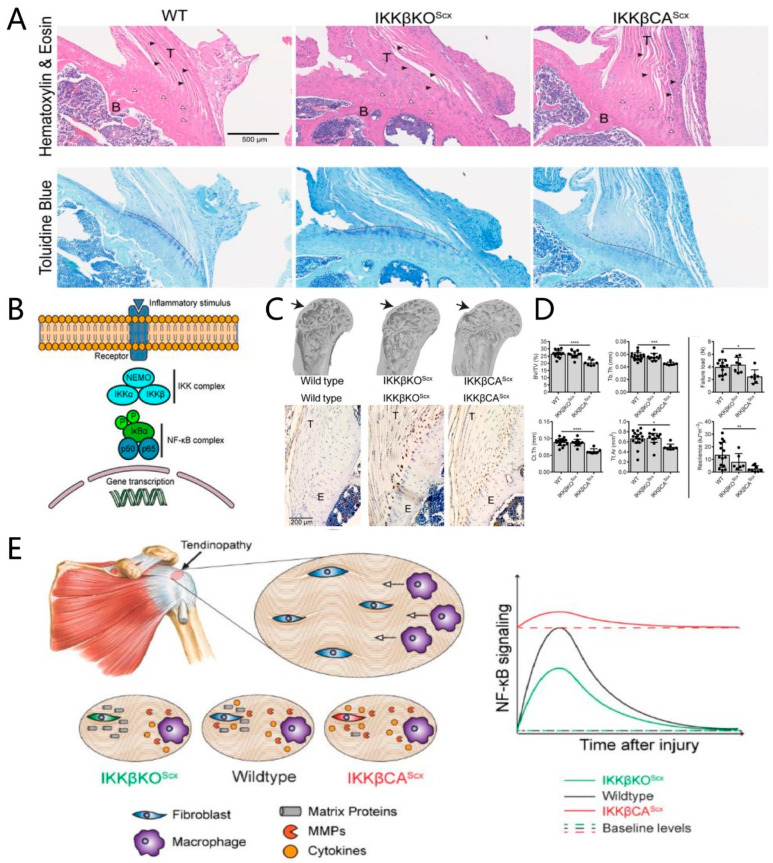
Activated IKK/NF-κB is detrimental to the tendon-to-bone interface and impedes tendon-bone healing. (**A**) Toluidine blue stain shows the loss of metachromasia at the fibrocartilage layer of the tendon-to-bone interface. Black arrowheads: spindle shaped tendon fibroblasts, white arrowheads: enthesis chondrocytes. Metachromasia demonstrating fibrocartilage interface can be seen below the dashed line in Toluidine blue stained sections; (**B**) schematic of NF-κB signaling and gene transcription; (**C**) microcomputed tomography (μCT) three-dimensional reconstruction shows the bone loss in activated IKK/NF-κB mice, while immunolabeling for CD68 shows more macrophage infiltration in activated IKK/NF-κB mice; (**D**) the quantification of bone morphometry and the quantification of mechanical properties show that IKK/NF-κB is detrimental to the tendon-to-bone interface. **** *p* < 0.0001, *** *p* < 0.001, ** *p* < 0.01, * *p* < 0.05; (**E**) schematic of how IKK/NF-κB drives chronic tendinopathy and impairs tendon-bone healing. WT: wild-type; IKKβKO^Scx^: IκB kinase, knocked out in the tendon fibroblast; IKKβCA^Scx^: IκB kinase, overexpressed in the tendon fibroblast; T: tendon; B: bone; E: enthesis (tendon-to-bone interface). Copyright 2019, The American Association for the Advancement of Science.

**Figure 5 jfb-13-00243-f005:**
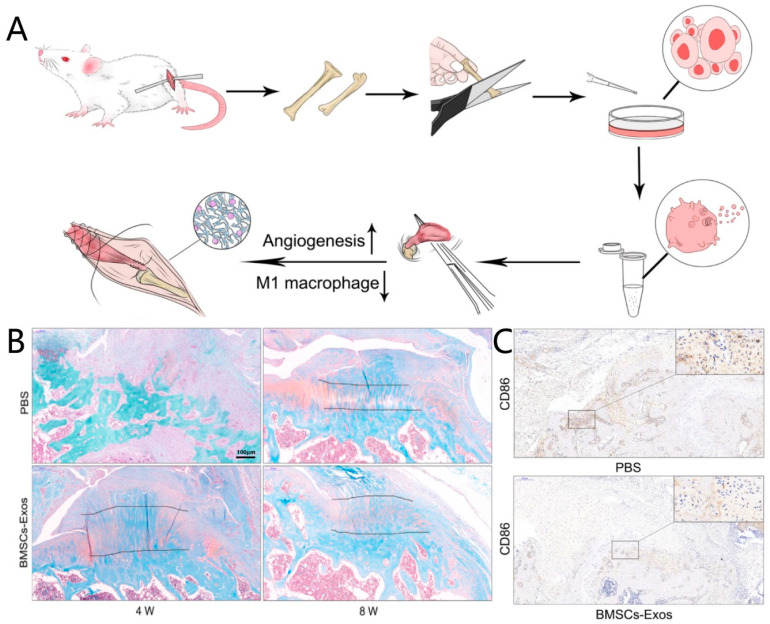
BMSC-Exos promotes tendon-bone healing after rotator-cuff reconstruction by promoting angiogenesis and inhibiting the M1 macrophage in rats. (**A**) The schematic diagram of these studies; (**B**) more fibrocartilage regeneration in the tendon-to-bone interface in the BMSC-Exos group; (**C**) less M1 macrophage marker CD86 in the tendon-to-bone interface in the BMSC-Exos group. PBS: phosphate-buffered saline; BMSCs-Exo: bone marrow stem cells exosome. Copyright 2020, Stem Cell Research & Therapy.

**Figure 6 jfb-13-00243-f006:**
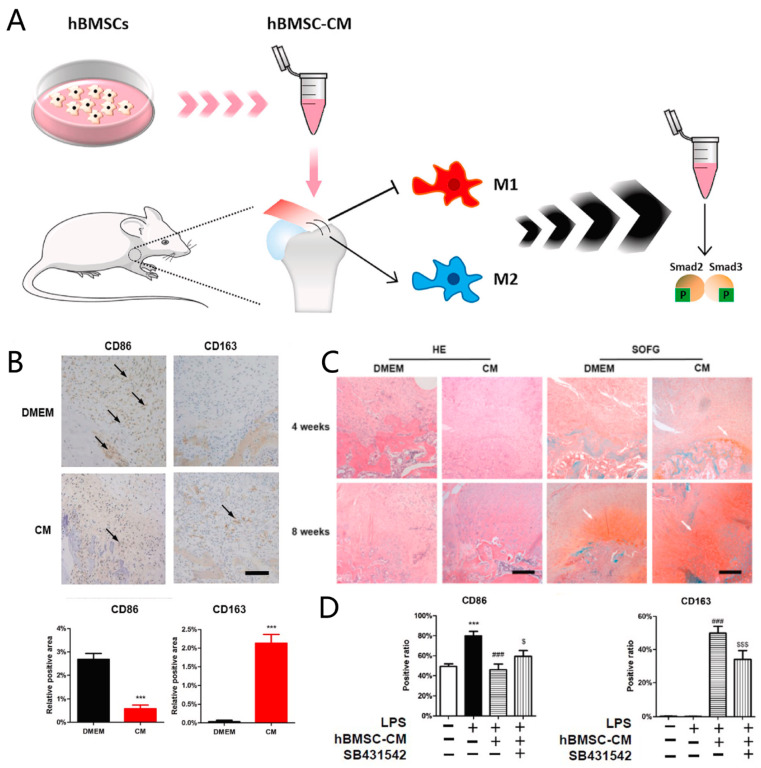
The conditioned medium of human bone marrow-derived stem cells (hBMSC-CM) promotes tendon-bone healing in a rat rotator cuff from a repair model. (**A**) The schematic diagram of this study; (**B**) There are fewer M1 macrophages and more M2 macrophages in the tendon-to-bone interface in the hBMSC-CM group than in the DMEM group. ***: *p* < 0.001 compared to DMEM group; (**C**) hBMSC-CM is beneficial for fibrocartilage regeneration; (**D**) hBMSC-CM regulates macrophage polarization through smad2/3 phosphorylation. ***: *p* < 0.001 compared to BMDMs without any stimulations. ###: *p* < 0.001 compared to BMDMs with LPS stimulation. $: *p* < 0.05 compared to BMDMs with LPS plus hBMSC-CM stimulation. $$$: *p* < 0.001 compared to BMDMs with LPS plus hBMSC-CM stimulation. hBMSC: human bone marrow stem cell; hBMSC-CM: human bone marrow stem cell conditioned medium; DMEM: Dulbecco’s modified Eagle medium; HE: hematoxylin-eosin staining; SOFG: Safranin O/Fast Green; LPS: lipopolysaccharide; SB431542: inhibitor of Smad2/3 phosphorylation; BMDM: Bone-Marrow Derived Macrophage. Scale bar 200μm. Copyright 2021, Elsevier.

**Figure 7 jfb-13-00243-f007:**
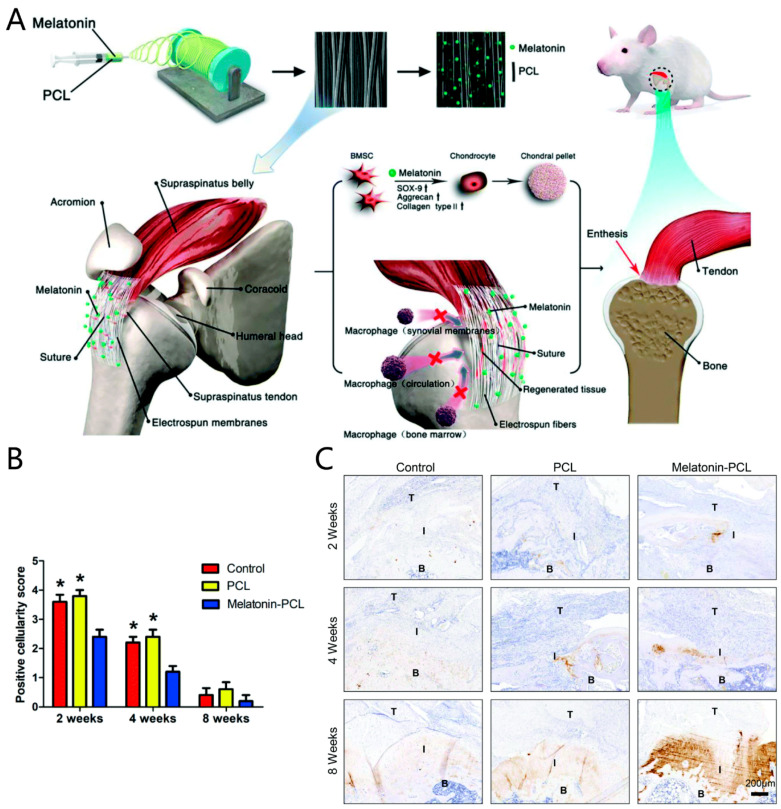
Pro-chondrogenic and immunomodulatory melatonin-loaded electrospun membranes for tendon-to-bone healing. (**A**) Illustration of melatonin-loaded PCL electrospun membranes for regenerating the tendon-to-bone interface and promoting tendon-to-bone healing in a rat rotator cuff tear model; (**B**) melatonin-loaded PCL electrospun membranes can decrease the infiltration of macrophages. *: significant difference between the melatonin-PCL group and the control or PCL group. *p* < 0.05; (**C**) melatonin-loaded PCL electrospun membranes promote fibrocartilage regeneration. PCL: polycaprolactone; T: tendon; I: tendon-to-bone interface; B: bone. Copyright 2019, Royal Society of Chemistry.

**Figure 8 jfb-13-00243-f008:**
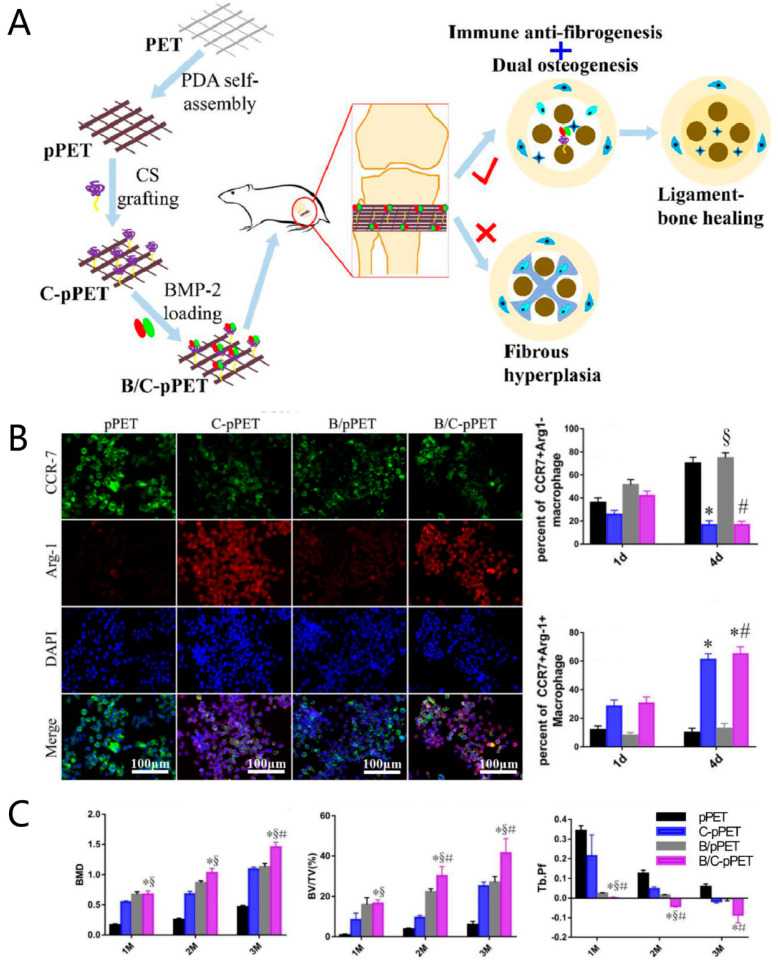
A triple-nano-coating polyethylene terephthalate graft containing chondroitin sulfate and bone morphogenetic protein-2 is used to facilitate ligament–bone healing. (**A**) The schematic diagram of this study; (**B**) chondroitin sulfate can promote M2 macrophage polarization; (**C**) CS and BMP-2 have a synergistic effect on bone regeneration. * indicates *p* < 0.05 compared to the pPET group; § indicates *p* < 0.05 compared to the C-pPET group; # indicates *p* < 0.05 compared to the B/pPET group. PET: polycaprolactone terephthalate; PDA: polydopamine; pPET: polydopamine-modified PET; C-pPET: chondroitin sulfate/polydopamine-modified PET; B/pPET: bone morphogenetic protein-2/polydopamine-modified PET; B/C-pPET: chondroitin sulfate/bone morphogenetic protein-2/polydopamine- modified PET. Copyright 2020, Elsevier.

**Table 1 jfb-13-00243-t001:** Strategies of modulating macrophages to promote tendon-bone healing.

Category	Method	Effective Component	Animal	Sites	Effects	References
Exosomes and secretome	Inject exosomes into the bone tunnel	Infrapatellar fat pad MSC-derived exosomes	Rats	ACL reconstruction of bone tunnel	Fewer M1 macrophages and more M2 macrophages; promote fibrocartilage and bone regeneration	[81]
	Hydrogel-loaded exosomes	Bone marrow MSC-derived exosomes	Mice	Achilles tendon	Higher numbers of M2 macrophages; more fibrocartilage regeneration; improves maximum force	[82]
	Inject exosomes into the tail vein	Bone marrow MSC-derived exosomes	Rats	Rotator cuff	Reduces the level of pro-inflammatory factors; increases the breaking load and stiffness	[77]
	Gelatin sponge loaded conditioned medium	Human MSCs-derived conditioned medium	Rats	Rotator cuff	Induces M2 macrophages polarization; promotes tendon-bone healing	[83]
Immunomodulatory molecules	Electrospun	Melatonin	Rats	Rotator cuff	Inhibits macrophage infiltration; increases chondroid zone formation; promotes collagen maturation	[87]
	Fibrin gel loaded acetylcholine and pyridostigmine	Acetylcholine and pyridostigmine	Mice	Rotator cuff	More M2 macrophages retain; failure load and stiffness are significantly improved	[88]
	Magnesium-pretreated periosteum	Magnesium	Rabbits	ACL reconstruction bone tunnel	Enhances M2 macrophage polarization; increases the formation of fibrocartilage; prevents peri-tunnel bone loss	[89]
	Triple-coated ligament graft	Chondroitin sulfate and BMP-2	Rats	Proximal-tibia bone tunnel	Regulates the polarization of macrophages; enhances new bone formation	[90]

## Data Availability

Not applicable.

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
