# Peer review of "Regulating Macrophages through Immunomodulatory Biomaterials Is a Promising Strategy for Promoting Tendon-Bone Healing"

_jfb, 2022, doi:10.3390/jfb13040243_

Round 1

Reviewer 1 Report

The current article under review summarizes the role of macrophages in tendon-to-bone healing following injury. Overall, the article is well structured and has highlighted recent developments in the field. The schematic figures in the introduction section are well made and self-explanatory.

I recommend that the authors proofread the manuscript to rectify any typographical errors. The only comment I have is to check the elastic modulus of the tendon mentioned in line 38. It reads 200 mPa but it should be MPa. As for the typographical errors, here is a non-exhaustive list- line 20 “paly”; line 178 “enhence”; figure 3 “ECM remodling.” 

Author Response

Point 1: The current article under review summarizes the role of macrophages in tendon-to-bone healing following injury. Overall, the article is well structured and has highlighted recent developments in the field. The schematic figures in the introduction section are well made and self-explanatory.

Response 1: Thank you for your comment.

Point 2: I recommend that the authors proofread the manuscript to rectify any typographical errors. The only comment I have is to check the elastic modulus of the tendon mentioned in line 38. It reads 200 mPa but it should be MPa. As for the typographical errors, here is a non-exhaustive list- line 20 “paly”; line 178 “enhence”; figure 3 “ECM remodling.” 

Response 2: Thank you for your constructive suggestion. We have corrected typographical errors and revised the manuscript according to your suggestion.

Reviewer 2 Report

The paper: “Regulating Macrophages through Immunomodulatory Bio- 2 materials is A Promising Strategy for Promoting Tendon-Bone healing” is an interesting review about the role of immune system in the regenerative system of tendon-bone disease, in particular the authors investigated the impact of different macrophages on tendon-bone healing, considering both challenges and limitations. The review is well written and detailed. The bibliography is well edited. However, I ask the authors to check all the paper and specify clearly the animal model used (when the experiments were not conducted on humans), this is a simple minor revision that will add  more value to the article.

Author Response

Point 1: The paper: “Regulating Macrophages through Immunomodulatory Biomaterials is A Promising Strategy for Promoting Tendon-Bone healing” is an interesting review about the role of immune system in the regenerative system of tendon-bone disease, in particular the authors investigated the impact of different macrophages on tendon-bone healing, considering both challenges and limitations. The review is well written and detailed. The bibliography is well edited. However, I ask the authors to check all the paper and specify clearly the animal model used (when the experiments were not conducted on humans), this is a simple minor revision that will add more value to the article.

Response 1: Thank you for your important suggestion. We have added the animal model description in the manuscript according to your suggestion.

Reviewer 3 Report

This manuscript is review discussing the bone and tendon healing strategy. The strategy involves use of biomaterials to modulate immune response of macrophages.  The content include mechanism of tendon and bone healing, macrophages role in bone and tendon healing, and types of macrophages involved in the healing process. Later, biomaterials such as exosomes, secretomes, immunomodulators, and physical therapy is discussed. 109 article are covered in this review, and text is supported by one table and eight figures.

In general, manuscript presents the interesting topic, and I believe it will be interesting for reader. However, it requires some corrections and improvements.

I suggest revising the title to improve clarity. Further, section ‘4.3 physical therapy’ looks out of the manuscript scope considering title, and outlier for section 4, considering the section heading. Please remove that section or revise title or give separate section for physical therapies to modulate macrophages. This would make it better aligned and clearer to readers.  

Please check spelling errors. For example “Macrophages paly” in abstract.  

Figures: Expansions to Shorts forms used in figures must be added in figure caption. Some figures are not readable. Statistics labels in bar graph are not presented in caption. In this case, readers will feel difficulty in understanding without referring to main articles.

Author Response

Point 1: I suggest revising the title to improve clarity. Further, section ‘4.3 physical therapy’ looks out of the manuscript scope considering title, and outlier for section 4, considering the section heading. Please remove that section or revise title or give separate section for physical therapies to modulate macrophages. This would make it better aligned and clearer to readers.

Response 1: Thank you for your constructive suggestion. We have removed ‘4.3 physical therapy’ section according to your suggestion.

Point 2: Please check spelling errors. For example “Macrophages paly” in abstract.  

Response 2: Thank you for your important suggestion. We have corrected spelling errors and revised the manuscript according to your suggestion.

Point 3: Figures: Expansions to Shorts forms used in figures must be added in figure caption. Some figures are not readable. Statistics labels in bar graph are not presented in caption. In this case, readers will feel difficulty in understanding without referring to main articles.

Response 3: Thank you for your critical suggestion. Shorts forms and statistics labels have been added in figure caption. We have added statistics labels and revised figures according to your suggestion.
